# Genomic insights into the conservation status of the world's last remaining Sumatran rhinoceros populations

Johanna von Seth [1,2,3,21]✉, Nicolas Dussex [1,2,3,21]✉, David Díez-del-Molino [1,2,3], Tom van der Valk [1,2,4], Verena E. Kutschera [5], Marcin Kierczak [6], Cynthia C. Steiner [7], Shanlin Liu[8], M. Thomas P. Gilbert [8,9], Mikkel-Holger S. Sinding [8,10], Stefan Prost [11,12], Katerina Guschanski [4,13], Senthilvel K. S. S. Nathan[14], Selina Brace [15], Yvonne L. Chan[1,2], Christopher W. Wheat[3], Pontus Skoglund[16], Oliver A. Ryder[7], Benoit Goossens[14,17,18,19], Anders Götherström[1,20] & Love Dalén [1,2,3]✉

Small populations are often exposed to high inbreeding and mutational load that can increase the risk of extinction. The Sumatran rhinoceros was widespread in Southeast Asia, but is now restricted to small and isolated populations on Sumatra and Borneo, and most likely extinct on the Malay Peninsula. Here, we analyse 5 historical and 16 modern genomes from these populations to investigate the genomic consequences of the recent decline, such as increased inbreeding and mutational load. We find that the Malay Peninsula population experienced increased inbreeding shortly before extirpation, which possibly was accompanied by purging. The populations on Sumatra and Borneo instead show low inbreeding, but high mutational load. The currently small population sizes may thus in the near future lead to inbreeding depression. Moreover, we find little evidence for differences in local adaptation among populations, suggesting that future inbreeding depression could potentially be mitigated by assisted gene flow among populations.

[1] Centre for Palaeogenetics, Stockholm, Sweden. [2] Department of Bioinformatics and Genetics, Swedish Museum of Natural History, Stockholm, Sweden. [3] Department of Zoology, Stockholm University, Stockholm, Sweden. [4] Department of Ecology and Genetics, Animal Ecology, Uppsala University, Uppsala, Sweden. [5] Department of Biochemistry and Biophysics, National Bioinformatics Infrastructure Sweden, Science for Life Laboratory, Stockholm University, Solna, Sweden. [6] Department of Cell and Molecular Biology, National Bioinformatics Infrastructure Sweden, Science for Life Laboratory, Uppsala University, Uppsala, Sweden. [7] San Diego Zoo Wildlife Alliance, Beckman Center for Conservation Research, Escondido, CA, USA. [8] The GLOBE Institute, University of Copenhagen, Copenhagen, Denmark. [9] Norwegian University of Science and Technology, University Museum, Trondheim, Norway. [10] Smurfit Institute of Genetics, Trinity College Dublin, Dublin, Ireland. [11] LOEWE-Centre for Translational Biodiversity Genomics, Senckenberg, Frankfurt, Germany. [12] South African National Biodiversity Institute, National Zoological Garden, Pretoria, South Africa. [13] Institute of Evolutionary Biology, School of Biological Sciences, University of Edinburgh, Edinburgh, UK. [14] Sabah Wildlife Department, Kota Kinabalu, Sabah, Malaysia. [15] Department of Earth Sciences, Natural History Museum, London, UK. [16] Francis Crick Institute, London, UK. [17] Organisms and Environment Division, Cardiff School of Biosciences, Cardiff, UK. [18] Sustainable Places Research Institute, Cardiff University, Cardiff, UK. [19] Danau Girang Field Centre, c/o Sabah Wildlife Department, Kota Kinabalu, Sabah, Malaysia. [20] Department of Archaeology and Classical Studies, Stockholm University, Stockholm, Sweden. [21] These authors contributed equally: Johanna von Seth, Nicolas Dussex. ✉email: johanna.n.vonseth@gmail.com; nicolas.dussex@gmail.com; love.dalen@nrm.se

Small and fragmented populations are vulnerable to several extrinsic and intrinsic threats, such as environmental effects (e.g., disease, habitat destruction), demographic stochasticity and detrimental genetic effects[1]. Over the past few decades, the role of genetic factors in the long-term viability of small populations has gained considerable attention[2,3]. A growing body of empirical studies on critically endangered species[4,5] indicates that small populations are often exposed to genomic erosion, which reduces species viability via loss of genetic diversity, increase in inbreeding and in genetic load (i.e., decrease in average individual fitness relative to the fittest genotype due to deleterious mutations) through genetic drift[6–8].

The magnitude of these processes can vary among populations and species due to their different demographic histories (e.g., population fluctuations and founder effects), sensitivity to environmental changes or life-history traits. Moreover, population fitness and viability will also vary depending on the proportion of rare large-effect deleterious alleles and numerous small-effect deleterious alleles among the founding individuals of these populations[9]. Predicting the degree of genomic erosion that threatened populations are exposed to is thus challenging, but has important implications for conservation, since quantifying levels of inbreeding and genetic load is crucial for developing management strategies aimed at mitigating the effects of genomic erosion[9].

Genetic data can, for instance, help define management units[10], contribute to breeding programmes by estimating relatedness among individuals and help identify the individuals most likely to lead to genetic rescue in translocation programmes[11]. Enhancing gene flow is considered a powerful conservation tool for reducing genomic erosion in a range of threatened species, because it can reduce the expression of recessive or partially-recessive deleterious alleles in hybrids of the receiving population[11,12]. There is empirical evidence for genetic rescue resulting in increased fitness, population growth, and increased heterozygosity in the first two or three hybrid generations[13–16]. Furthermore, while genomic data on the long-term effects of gene flow are currently limited, hybrid vigour can persist at least beyond the F3 generation and as far as the F16 generation[17–20].

Populations that have evolved in isolation for thousands of generations may show evidence for positive selection and distinct allelic frequencies for genes under strong directional selection, which could indicate local adaptation[21]. Thus, assisted gene flow among long-term isolated populations represents a risk of disrupting locally adapted gene complexes, a process referred to as outbreeding depression[3,6]. Previous work has highlighted factors likely to affect the risk of outbreeding depression, where adaptive differentiation has been identified as one of the major risk factors[22]. Indeed, concerns over potential disruption of local adaptation has been one of the main reasons why assisted gene flow has only rarely been used in conservation biology (~34 studies[19]).

While populations can become locally adapted[23], they may also accumulate different levels of genetic load, and potentially carry private deleterious mutations that affect different genes. Consequently, gene flow from outbred populations into isolated populations could increase the genetic load of recipient populations and elevate their risk of extinction[24,25]. This is especially likely if the recipient population has purged a portion of its genetic load but remains small and inbred for several generations. Such a population will be vulnerable to the expression of deleterious alleles in homozygous state, and the effects of newly introduced deleterious alleles may not become apparent until several generations later[24,26]. In such situations, it could thus be preferable to avoid genetic rescue attempts or to select individuals with a lower genetic load from partially-inbred sources[24,25].

However, some have criticised this approach and argue that the benefits of genetic rescue (i.e., hybrid vigour and maintenance of adaptive potential) by far outweigh the risks of increase in genetic load[27]. It is therefore essential to weigh the positive and negative effects of this alternative approach when assessing the need for genetic rescue[27].

Recent advances in genomics have allowed the detection of genomic regions affected by natural selection and can also help identify genetic threats such as elevated inbreeding and genetic load in natural populations[28]. However, estimating genetic load is challenging without information on the fitness effects of deleterious mutations. An alternative approach is to estimate changes in mutational load (i.e., number of deleterious mutations), and to use it as a proxy for individual and population fitness. Determining individual inbreeding levels as well as mutational load can thus potentially be used to identify individuals particularly well suited for assisted gene flow.

The Sumatran rhinoceros (Dicerorhinus sumatrensis) is a browser that inhabits the rainforest of Southeast (SE) Asia and is currently listed as critically endangered by the International Union for Conservation of Nature (IUCN)[29]. Mitogenomic and genome-wide data indicate that periods of sea-level rise during the Pleistocene led to repeated isolation among populations[30,31]. A recent study using complete mitochondrial genomes estimated the divergence of the three main Sumatran rhinoceros populations at ~360 ka BP[31] and additional substructure within the three main populations are roughly coinciding with the Toba super-eruption some 71 ka BP[32], which may have restricted Sumatran rhinoceros populations to refugia after the eruption. At the end of the last glaciation, the species experienced a severe decline in effective population size ($N_e$), which may have eroded a large portion of its ancestral genetic diversity[30].

Sumatran rhinoceroses were until recently widespread in SE Asia, from as far as the foothills of the Himalayas or Assam down to the islands of Sumatra and Borneo[33,34]. It has been estimated that the census population size has decreased by ~70% over the past 20 years as a result of poaching and habitat change, but population declines had already been reported in 1939[31,34–37]. However, it is difficult to obtain accurate estimates of the historical and contemporary population sizes since Sumatran rhinoceroses are solitary and live in dense rainforests. As of 2019, the species likely numbers fewer than 100 individuals and only small, fragmented populations survive on Sumatra (D. s. sumatrensis) and Borneo (D. s. harrissoni), whereas the Malay Peninsula (D. s. sumatrensis) population is most likely extinct[38,39]. Moreover, the species' low breeding rate, in the wild due to low population density and due to female reproductive pathologies in captivity, makes it one of the most endangered rhinoceros species in the world[40]. On Borneo, the situation is especially dire since only one female survives in captivity and < 10 wild individuals remain in East Kalimantan[41].

Consistent with the historical demographic decline since the 1930s, corresponding to approximately seven generations, the Sumatran rhinoceros is thought to have experienced a loss of genetic diversity both in the wild and in captivity[35,36]. Consequently, the remaining populations may be exposed to intrinsic genetic threats such as reduced adaptive potential and inbreeding depression, which could accelerate its decline[2,3].

With fewer than 100 individuals remaining, there have been recent proposals to manage the species as a single unit and to increase gene flow by translocation or exchange of gametes from different populations[40]. However, the differentiation of these three genetic lineages possibly indicates a risk for outbreeding depression as a result of mixing populations, which implies that there may be a need to treat Sumatran rhinoceros subspecies as different conservation units (e.g., Steiner et al.[31]). The risks of

outbreeding depression may be outweighed by the risk of extinction due to inbreeding, demographic stochasticity, or environmental effects such as disease[31], but concerns about outbreeding depression and the introduction of new deleterious alleles have not yet been addressed[31,35,36]. Thus, from a genetic perspective, the management of the remaining Sumatran rhinoceros populations represents a conundrum to conservation biologists, where the risks (i.e., introduction of maladapted or deleterious alleles) have to be weighed against the benefits (i.e., increase of population fitness via genetic rescue).

Here, we aim to gain insights into the effects of population decline on the genetic conservation status of Sumatran rhinoceros by examining five historical and 16 modern whole-genomes from the two remaining populations on Sumatra and Borneo, and the most likely extinct population on the Malay Peninsula. We first investigate the population structure, past demography and timing of divergence among these three populations. Second, we estimate differences in genomic diversity and mutational load among the three populations, as well as temporal changes within two of the populations (Borneo and Malay Peninsula) using museum samples up to 140 years old. Finally, to evaluate the potential risks associated with assisted gene flow among modern populations, we examine the extent of private mutational load within each population, and whether signatures of positive selection, potentially associated with local adaptation, differ among the populations.

## Results

**Population structure and demographic history.** In order to estimate genomic diversity and mutational load in modern and historical populations, we mapped paired-end data from 18 re-sequenced genomes (four historical and 14 modern) Sumatran rhinoceros specimens from Sumatra, Borneo and the Malay Peninsula to a de novo assembly reference genome for Sumatran rhinoceros (Supplementary Table 1, Genbank: GCA_014189135.1. Genome coverage ranged from 9X to 29X (mean: 19X; Supplementary Table 2; see 'Methods'). For population structure analyses we also included three additional genomes that had lower coverage (mean: 3X).

Consistent with previous analyses of mitogenome data[31], our phylogenetic tree based on pairwise genetic distances and principal component analysis (PCA) revealed three distinct and reciprocally monophyletic clusters corresponding to the Sumatran, Malay Peninsula and Bornean populations (Fig. 1b, Supplementary Fig. 1, Supplementary Table 1; see 'Methods'). Interestingly, the phylogenetic tree and a clustering analysis showed further distinction between two lineages within the island of Sumatra, roughly corresponding to the northeastern Sumatra and southwestern Sumatra clades described in Steiner et al.[31] (Supplementary Fig. 2, Supplementary Table 3; see 'Methods'). However, in contrast to Steiner et al.[31], all individuals from the Malay Peninsula grouped together and formed a sister lineage to the Sumatran lineage.

Using the PSMC[42], we estimated temporal fluctuations in $N_e$ for the Sumatran rhinoceros (see 'Methods'). All three populations experienced similar demographic trajectories characterised by a decrease in $N_e$, with the most severe decline starting some 700–500 ka BP and by a subsequent more gradual decline to a $N_e$ of ~2,000–1,000 at the end of the last glaciation (Fig. 2). Although there was some individual variation in the amplitude of these trajectories, the declines were consistent among all individuals of the three populations (Supplementary Figs. 3 and 4). However, our analyses also revealed one important difference among the three populations. The Sumatran and Malay Peninsula populations showed nearly identical demographic trajectories characterised by a slight recovery ca. 200–100 ka BP and population stability in the following 60 ka. The Bornean population instead continued to decline after 200 ka BP, but might have experienced some population stability around 60–20 ka BP.

To further investigate Sumatran rhinoceros demography, we investigated gene flow and split times ($T$) among the three populations using pseudodiploid X chromosome genomes for each pair of populations (see 'Methods')[42,43]. The PSMC indicated that the Bornean population started diverging from the Sumatran and Malay Peninsula populations ~300 ka BP (Fig. 2, Supplementary Fig. 5, Supplementary Table 4). However, the curves showed several steps between 300 ka and 30–50 ka BP, when $N_e$ finally reached infinity, indicating a gradual isolation of the Bornean population. In contrast, the divergence among the populations on the Malay Peninsula and Sumatra occurred much later, sometime between 9 and 13 ka BP, with $N_e$ reaching infinity between 6 and 3 ka BP (Supplementary Fig. 5).

**Genetic diversity and inbreeding in modern-day populations.** We estimated the Sumatran rhinoceros' spatial differences in genome-wide heterozygosity ($\theta$) measured as the number of heterozygous sites per 1000 bp[44] and inbreeding as the proportion of the genome found in runs of homozygosity[45,46] ($F_{ROH}$) using 4,656,534 high-quality SNP calls (see 'Methods').

We found significant differences in heterozygosity and inbreeding ($F_{ROH} \geq 2$ Mb) among populations, with Sumatra and Borneo having ~71 and 78% higher heterozygosity ($\theta_{Sumatra} = 1.42$; $\theta_{Borneo} = 1.48$; t-test, $p = 4.67e{-}07$; Supplementary Fig. 6a), as well as ~71 and 85% lower inbreeding levels ($F_{ROH\text{-}Sumatra} = 0.086$; $F_{ROH\text{-}Borneo} = 0.045$; t-test, $p = 1.03e{-}05$) compared to the recently extinct Malay Peninsula population ($\theta_{MalayP.} = 0.83$; $F_{ROH\text{-}MalayP.} = 0.30$; Fig. 3a, Supplementary Figs. 6a–9, Supplementary Table 5). While the Bornean and Sumatran populations have on average < 10% of the genome contained within long ($F_{ROH} \geq 2$ Mb) ROH segments, the Malay Peninsula population had 30% shortly prior to its extinction.

**Mutational load in modern-day populations.** Because genetic load is challenging to estimate without any information on the fitness effects of deleterious variants, we here focus on quantifying differences in mutational load (i.e., number of potentially deleterious variants). We estimated mutational load using two complementary approaches. First, we estimated individual relative mutational load as the number of homozygous and heterozygous derived alleles at sites that are under strict evolutionary constraints using genomic evolutionary rate profiling scores (i.e., GERP scores; see 'Methods'). Genomic sites that have been strongly conserved for millions of years of evolution (i.e., across several taxonomic groups) are expected to be functionally important, and thus the fraction of mutations at such sites can serve as a proxy for mutational load (Supplementary Figs. 11 and 12, Supplementary Note 1)[47,48]. The resulting GERP scores indicated a significantly higher relative mutational load on Borneo compared to Sumatra (t-test, $p = 0.001$) as well as Malay Peninsula (t-test, $p = 0.0001$), and a significantly higher relative mutational load on Sumatra compared to Malay Peninsula (t-test, $p = 0.012$, Fig. 3b, Supplementary Fig. 13). Moreover, we found that 18% of deleterious alleles in highly conserved regions (i.e., GERP-score > 4) are shared among the three populations, whereas 30%, 20% and 14% are unique to the Bornean, Sumatran and Malay Peninsula populations, respectively (Supplementary Table 6).

Second, we estimated mutational load in coding regions using an annotation of 33,026 genes, based on the white

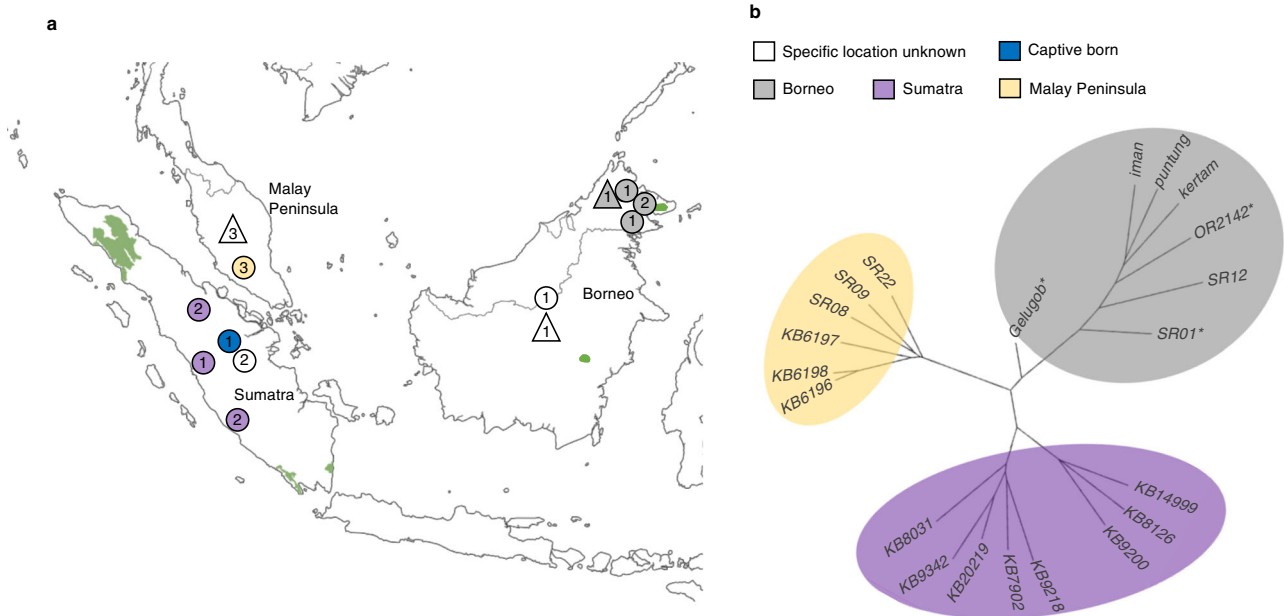

**Fig. 1 Sampling, current range, and phylogeny for Sumatran rhinoceros (*Dicerorhinus sumatrensis*). a** Geographical origin of the 21 Sumatran rhinoceros genomes analysed in this study. Current distribution is depicted in green[40]. Triangles and circles represent approximate geographical sampling locations of historical and modern samples, respectively. The numbers within the geometric symbols depict the number of samples from the corresponding location. **b** Unrooted phylogeny (100 bootstrap replicates). Asterisks denote low coverage (< 9X) genomes.

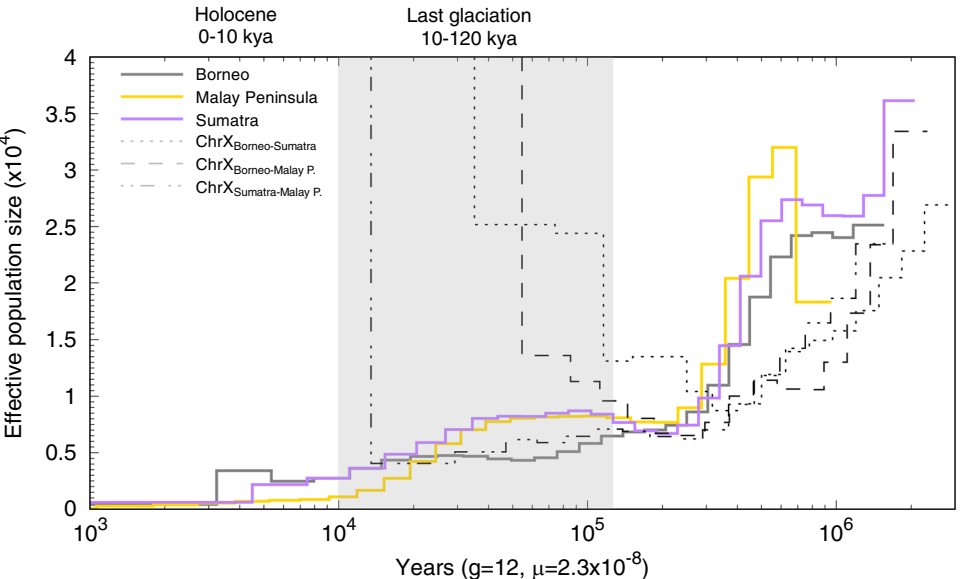

**Fig. 2 Population history and timing of population divergence for Sumatran rhinoceros (*Dicerorhinus sumatrensis*).** Thick coloured lines depict temporal fluctuations in effective population size ($N_e$) for the three populations with each coloured line representing one individual from each population. Dotted/dashed black curves represent the pseudodiploid sex chromosomes (i.e., X chromosome) used to infer pairwise population divergence times, with the curves going to infinity at the respective time of divergence. The X-axis corresponds to time before present in years on a log scale, assuming a substitution rate ($\mu$) of $2.34 \times 10^{-8}$ substitutions/site/generation[30,82] and a generation time ($g$) of 12 years[76]. The y-axis corresponds to $N_e$. The grey rectangle depicts the last glaciation. Bootstrap tests were conducted with 100 replicates (Supplementary Fig. 4).

rhinoceros genome (*Ceratotherium simum simum*; Genbank: GCF_000283155.1 see 'Methods'). We identified sites carrying non-synonymous loss-of-function (LoF) variants, which are likely to be deleterious with a disruptive impact on protein function, and compiled a list of genes with such LoF variants. In total, we found 373 LoF variant sites across all three populations (Supplementary Table 7), that individuals on Borneo have a significantly higher

number of LoF variants ($n_{average} = 213$) compared to the other two modern populations (*t*-test, $p = 2.78e{-}05$), and that individuals on Sumatra have a significantly higher number of LoF variants ($n_{average} = 186$) compared to Malay Peninsula ($n_{average} = 165$; *t*-test, $p = 0.0019$, Fig. 3c). Significant differences among populations were only found for LoF variants in heterozygous state (*t*-test, $p = 3.21e{-}06$, Supplementary Fig. 14). In total, 335 genes

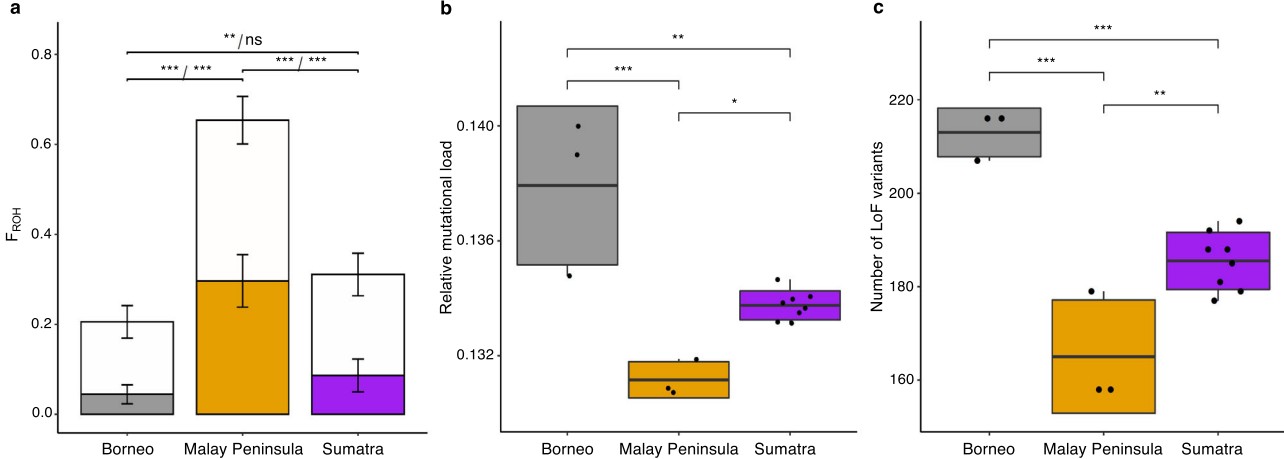

**Fig. 3 Inbreeding, relative mutational load and number of loss-of-function (LoF) variants in three modern populations of Sumatran rhinoceros (*Dicerorhinus sumatrensis*). a** Inbreeding estimated as the average proportion of the genome in runs of homozygosity ($F_{ROH}$). Open bars show total proportion of the genome in ROH ≥ 100 kb and solid bars show proportions in ROH of length ≥ 2 Mb. Bars extending from the mean values represent the standard deviation (two-sided pairwise *t*-test, $F_{ROH}$ ≥ 100 kb: $p_{Borneo-MalayP} = 4.2e−07$, $p_{Borneo-Sumatra} = 0.0066$, $p_{MalayP-Sumatra} = 6.4e−07$, $F_{ROH}$ ≥ 2 Mb: $p_{Borneo-MalayP} = 2.2e−05$, $p_{Borneo-Sumatra} = 0.15$, $p_{MalayP-Sumatra} = 2.2e−05$). **b** Mutational load estimated with GERP scores. Individual relative mutational load was measured as the sum of all derived alleles multiplied by their conservation-score over the total number of derived alleles (see 'Methods'). Only derived alleles above conservation-score of 1 (i.e., non-neutral) were included (two-sided pairwise *t*-test, $p_{Borneo-MalayP} = 0.00013$, $p_{Borneo-Sumatra} = 0.001$, $p_{MalayP-Sumatra} = 0.012$. **c** Number of LoF variants using an annotation of 33,026 gene predictions for a white rhinoceros genome assembly (see 'Methods') (two-sided pairwise *t*-test, $p_{Borneo-MalayP} = 2.2e−05$, $p_{Borneo-Sumatra} = 0.0004$, $p_{MalayP-Sumatra} = 0.0019$). Middle thick lines within boxplots and bounds of boxes represent mean and standard deviation, respectively. Vertical lines represent minima and maxima. $n = 14$, ***$p < 0.001$, **$p < 0.01$, *$p < 0.05$, ns = non-significant, *p*-values were not adjusted for multiple comparisons.

were found to carry LoF variants, and the majority (Sumatra: 191 out of 208, Borneo: 198 out of 219, and Malay Peninsula: 123 out of 136) of the identified genes had one LoF variant (Supplementary Table 8). The highest number of LoF variants per gene was six. There were private LoF variants in each population but only a small number of LoF variants were fixed in each population, with seven and 13 variants in the Bornean and Malay Peninsula populations, respectively, but none in the Sumatran population (Supplementary Table 7). However, only two of these fixed LoF variants were private for the population on Borneo, and none for the Malay Peninsula population (Supplementary Table 7). Nonetheless, interbreeding with the Sumatran population could result in the masking of the seven fixed variants for the Bornean population.

Out of the 335 genes carrying LoF variants, we found 77, 99 and 24 genes that were affected only in the Sumatran, Bornean and the Malay Peninsula populations, respectively (Supplementary Fig. 15, Supplementary Table 7). Gene Ontology (GO) analysis (see 'Methods') indicated that genes with LoF variants were associated with cellular, metabolic and developmental processes as well as immunity (e.g., MHC; Supplementary Table 9). However, there was no indication of any biological functions being statistically overrepresented among genes with LoF variants.

Translocations of single individuals are unlikely to lead to long-term genetic rescue effects in wild Sumatran rhinoceros populations. However, since the number of animals surviving in the wild is very small and because obtaining gametes for a large number of individuals may be challenging, we assessed the possible effects of gene flow from specific individuals in our dataset on the number of LoF variants. We found that one single individual could introduce an average of ten new LoF variants into any of the other populations (Supplementary Fig. 16). However, we observed a large inter-individual variation, meaning that depending on the choice of donor, as few as one and as many

as 75 new LoF variants could be introduced into the receiving population.

**Temporal changes in genetic parameters**. When examining temporal changes in genome-wide heterozygosity and inbreeding in the populations for which we obtained both modern and historical samples (i.e., Borneo and Malay Peninsula), we found that the Malay Peninsula population experienced a significant 1.5- and 3.84-fold increase in the proportion of all ROH ($F_{ROH}$ ≥ 100 kb: $F_{ROH-hist.} = 0.44$; $F_{ROH-mod.} = 0.65$; *t*-test, $p = 0.034$ and long ROH ($F_{ROH}$ ≥ 2 Mb: $F_{ROH-hist.} = 0.078$; $F_{ROH-mod.} = 0.30$; *t*-test, $p = 0.007$) during the past century, respectively (Fig. 4a, Supplementary Table 5). We observed no significant temporal change in $F_{ROH}$ in Borneo, or in $\theta$ in either of the populations (Supplementary Fig. 6b). Moreover, we found a temporal decrease in mutational load estimated with GERP scores for the Malay Peninsula population (Fig. 4b; *t*-test, $p = 0.047$) but no difference in the number of LoF variants per individual genome (Fig. 4c, Supplementary Fig. 17).

**Detection of positive selection as proxy for local adaptation**. We used two complementary approaches to test for positive selection potentially associated with local adaptation to different island habitats.

First, we identified non-synonymous variants classified as missense (i.e., non-disruptive variants that might change protein function and efficiency; see 'Methods') and estimated their frequencies in each population. We identified a total of 15,598 missense variants (excluding missense variants fixed in all Sumatran rhinoceroses), which were associated with 6,490 identifiable genes (Supplementary Table 10). There were 1,409, 1,505 and 379 genes that showed missense variants only in the Sumatra, Borneo and Malay Peninsula populations, respectively, but not in the other two populations (Supplementary Fig. 18).

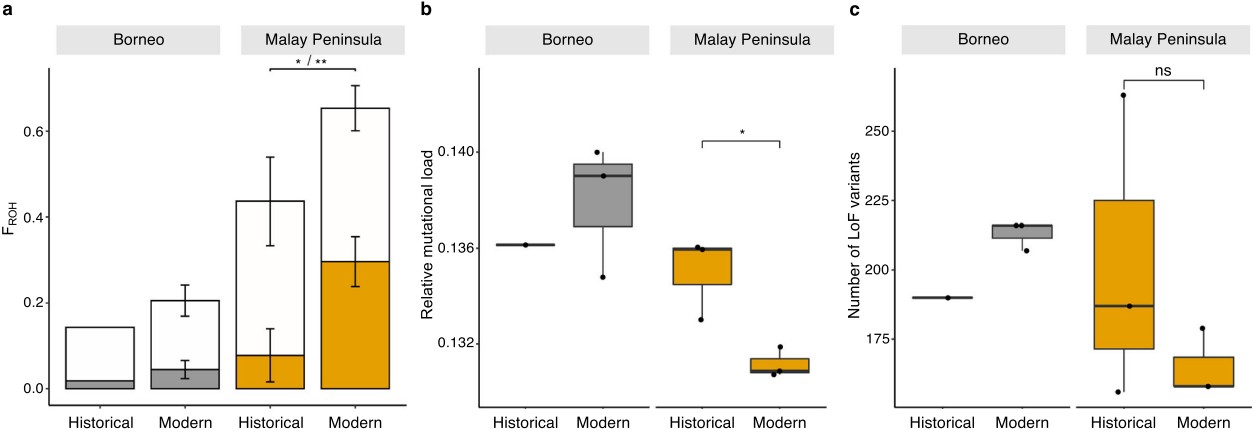

**Fig. 4 Temporal changes in inbreeding, relative mutational load and number of loss-of-function (LoF) variants in two populations of Sumatran rhinoceros (*Dicerorhinus sumatrensis*). a** Inbreeding estimated as the average proportion of the genome in runs of homozygosity ($F_{ROH}$). Open bars show the total proportion of the genome in ROH ≥100 kb and solid bars show proportions in ROH of length ≥ 2 Mb. Bars extending from the mean values represent the standard deviation (two-sided *t*-test, $F_{ROH}$ ≥ 100 kb: $p = 0.034$, $F_{ROH}$ ≥ 2 Mb: $p = 0.007$). **b** Mutational load estimated with GERP scores. Individual relative mutational load was measured as the sum of all derived alleles multiplied by their conservation-score over the total number of derived alleles (see 'Methods'). Only derived alleles above conservation-score of 1 (i.e., non-neutral) were included (two-sided *t*-test, $p = 0.047$). **c** Number of LoF variants using an annotation of 33,026 genes for white rhinoceros (see 'Methods') (two-sided *t*-test, $p = 0.35$). Middle thick lines within boxplots and bounds of boxes represent mean and standard deviation, respectively. Vertical lines represent minima and maxima. $n = 6$, **$p < 0.01$, *$p < 0.05$, ns = non-significant, *p*-values were not adjusted for multiple comparisons.

Among these, there were zero, 103 and 107 genes that had fixed variants in the Sumatra, Borneo and Malay Peninsula populations, respectively (Supplementary Table 10).

Next, we used population branch statistics (PBS), which incorporate the $F_{ST}$ distance between a population of interest and two other populations[49], to estimate the amount of divergence specific to the branch and for each gene (see 'Methods'). Out of 33,026 high-quality gene models, we found 61 genes with PBS values larger than the 99.8th percentile of the distribution of the PBS values of all genes (Supplementary Fig. 19, Supplementary Table 11). We found seven and two genes with a signal of positive selection unique to the Bornean and Malay Peninsula populations, respectively, and none in the Sumatran population (Supplementary Fig. 20). Twelve genes were identified with an infinite PBS value (i.e., $F_{ST} = 1$; Supplementary Table 11) over all pairwise population comparisons, indicating fixed allelic differences among our samples. Although this could indicate that the populations represent lineages that have evolved unique adaptive differences in these genes, we caution that analysis of additional individuals would be necessary to ascertain that these variants are truly fixed among the populations. Finally, out of the 61 genes detected as positively selected with PBS, two had missense variants (Supplementary Table 11).

Gene Ontology (GO) categories of all genes with missense variants ($n = 6,490$) and genes identified as under positive selection in the PBS analysis ($n = 61$) were related to various biological functions including cellular processes, biological regulation and metabolic processes (Supplementary Table 12). However, no GO categories were overrepresented in either population.

## Discussion

The Sumatran rhinoceros has inhabited Southeast Asia since the beginning of the Pleistocene[50,51]. Our demographic reconstructions show that the histories of the populations on Sumatra, Borneo and the Malay Peninsula are characterized by a continuous decline for several hundred thousand years. However, even though these three regions were connected during the Late

Pleistocene (ca. 110–12 ka BP) due to lowered sea levels[52], our analyses of pseudodiploid X chromosomes suggest that the Bornean population started to become isolated already 300 ka BP, which is consistent with previous estimates based on mitogenome data[31]. At that time, the Sunda shelf was not completely submerged, with land linking mainland Asia with Sumatra and Borneo[53]. However, even though these regions were connected, a savannah corridor could have formed a geographical barrier for east-west dispersal of rainforest species such as Sumatran rhinoceros[54], thereby limiting gene flow between Borneo and Sumatran/Malay Peninsula. In contrast, the populations on Sumatra and the Malay Peninsula appear to have become isolated from each other more recently, during the past 13–3 ka BP.

While it may be difficult to distinguish between population structure and population decline with the PSMC approach[55], the relatively small $N_e$ during the Holocene shown in the PSMC analysis, are consistent with a strong differentiation among the populations and with each population being reciprocally monophyletic. We also find evidence for a divergence between the northeastern and southwestern parts of Sumatra, most likely due to isolation caused by a mountain range stretching along the west coast of Sumatra. This subdivision is geographically different from the genetic divergence identified in Sumatran orangutans (*Pongo abelii* and *Pongo tapanuliensis*)[32,56], with an initial divergence among orangutan populations north and south of Lake Toba starting ~3.38 mya and gene flow ceasing as a result of a volcanic eruption some 71 ka BP[56]. Additional sampling of rhinoceros specimens from north of Lake Toba would be needed to investigate the possibility of a similar deep subdivision in the Sumatran rhinoceros.

Our PSMC analysis indicates that the Sumatran and Malay Peninsula lineages separated around 13 ka BP, whereas the Bornean and Sumatran lineages started separating much earlier, some 300 ka BP. Moreover, there are indications of body size differences among populations, with individuals from the Bornean population being smaller than individuals from other populations[57,58]. These size differences and our results are thus consistent with the current taxonomy and subspecies delimitation. The level of genetic divergence in Sumatran rhinoceros is

similar to what has recently been described for the white rhinoceros (*C. simum sp.*), where evidence for long-term divergence (80–10 ka BP) and limited post-divergence gene flow, as well as local adaptation, have been identified for the northern and southern subspecies[59]. However, it is worth noting that while long-term geographical isolation in Sumatran rhinoceros could lead to local adaptation, adaptive changes may not always occur, as indicated by our tests for positive selection in the three populations.

Consistent with the inferred small $N_e$ over the past 10 ka BP, we also observe a large number of short ROH (< 2 Mb) within all three populations. Such a pattern is expected in populations with a long-term small population size due to repeated random mating between distant relatives, but is also common in populations that have experienced historical bottlenecks, since recombination breaks down long ROH into smaller ones[60].

However, even though the Sumatran rhinoceros has gone through a major decline in the past century, to the extent that fewer than 100 individuals currently remain[39], we find relatively little evidence for recent inbreeding in the populations on Borneo and Sumatra. When estimating inbreeding levels based on ROH ≥ 2 Mb, we found that, on average, < 10% of the genomes contain longer ROH segments. In addition, individual genome-wide heterozygosity was higher than in some other endangered taxa (e.g., Amur tiger, crested ibis, Grauer's gorilla[9]), as well as in white rhinoceros[59]. While these comparatively low levels of inbreeding and high levels of genetic diversity imply that the two surviving populations have not yet been affected by strong inbreeding depression[3], even a small number of lethal equivalents could in theory lead to reductions in fitness[61,62]. Moreover, we note that there is a considerable amount of mutational load in these populations, especially on Borneo. In light of the current small census population sizes on Sumatra and Borneo, a future increase in inbreeding seems highly likely. This could, in turn, lead to increased exposure of recessive mutations in homozygous state[63], which could pose a serious threat to the long-term persistence of these populations.

In contrast to the extant populations, the recently extinct population on the Malay Peninsula had lower mutational load, but a higher proportion of the genome contained within longer ROH segments (30%). The comparatively low mutational load in the Malay Peninsula population could be a consequence of long-term small $N_e$, consistent with the observations of a long-term small $N_e$ in mountain gorillas[4] and island foxes[64], which could have favoured purging of strongly deleterious alleles. We find some support for this in the PSMC, which indicated a slightly smaller $N_e$ in the population over the past 30 ka in comparison to the two extant populations. Alternatively, since our temporal comparison with ca. 100-year-old samples from the Malay Peninsula indicates that inbreeding increased and mutational load decreased during the past century, the Malay Peninsula population may have experienced inbreeding depression during this period, which could have resulted in purging of mutational load prior to its extinction[65,66]. We caution here that even though we found a temporal decrease in mutational load, additional genomes from the Malay Peninsula population would be needed to fully resolve the timing and extent of temporal changes in mutational load in this population.

The recent discovery of a wild female on Borneo (Indonesian province of East Kalimantan) has intensified the discussions on assisted gene flow and the role of genetic rescue between Borneo and Sumatra in order to alleviate the negative consequences associated with small and isolated populations[40]. Moreover, since the death of the last known captive male (Kertam) on Borneo, artificial insemination from its frozen sperm has been proposed

as a promising tool to introduce new genetic diversity into the Sumatran population (John Payne, pers. comm.).

Gene flow can improve the genetic status of an endangered species and result in an increase in fitness, known as hybrid vigour, through the masking of deleterious alleles[12,16]. For the extant Sumatran rhinoceros populations, since most of the identified LoF variants were in heterozygous state, gene flow would probably not lead to a marked increase in the masking of deleterious alleles. However, if inbreeding increases in the near future, gene flow could counteract the increased risk for fixation of the existing LoF variants.

One of the major risks of assisted gene flow is the disruption of local adaptation in the receiving population, which can cause outbreeding depression[21,22]. While we found several genes with fixed missense variants in the populations on Borneo and Malay Peninsula, such mutations can also lead to a reduced protein effectiveness or even loss of protein function. Yet, it is worth noting that even if these missense variants have a negative effect on fitness, these would be masked in heterozygous state and at lower frequency in the recipient population, thereby reducing these potentially negative fitness effects. Nevertheless, it is unclear whether these variants represent any specific selective advantage in the local environment. Because of this, we also estimated population branch statistic (PBS) scores, which have been shown to have a high power to detect recent natural selection[49] and should not be confounded by deleterious mutations. The results suggest that seven, two and zero genes have been uniquely selected in the Bornean, Malay Peninsula and Sumatran populations, respectively.

Thus, we do not find strong support for local adaptation in these three populations, suggesting that the introduction of locally maladapted genes via gene flow from other populations is not very likely. The overall high genomic divergence between the two extant Sumatran rhinoceros populations instead suggests that gene flow could lead to an increase in adaptive potential by increasing the allelic diversity of the receiving population[16,67]. We caution, however, that behavioural changes in relation to captivity and/or translocation, or the risk of disease transmission, have not been considered in this study, but might also have an impact on the outcome of assisted gene flow. Despite the lack of evidence for local adaptation, long-term isolation between populations, with the estimated divergence between the Bornean population and the two other populations being at least 30 times older than the divergence between the Sumatra and the Malay Peninsula populations, suggests that the three populations could represent distinct evolutionary significant units (ESUs[68,69]). From the perspective of maintaining evolutionary processes, these should be ideally conserved as separate historically isolated ESUs[68,69]. However, given the high risk of extinction for the Sumatran rhinoceros, one could argue that the remaining populations should be managed as a metapopulation[40], since increasing the evolutionary potential and chances of survival for the species through genetic rescue might outweigh the value of maintaining distinct evolutionary lineages[12,70].

While there seems to be little risk of introducing locally maladapted alleles, a potential by-product of gene flow could be an introduction of new deleterious variants into the recipient population[25]. We found evidence for private mutational load in the form of specific genes having deleterious alleles in one population but not in the other two populations. Furthermore, our estimates suggested that gene flow of one individual from one population to the other would, on average, introduce ten new LoF variants into the recipient population. However, we also showed that the choice of donor would have a marked effect on how many deleterious alleles would be introduced, with estimates

ranging from one to 75 new LoF variants depending on which individual was selected for assisted gene flow. Similarly, there is an asymmetry in the cost and benefit of gene flow among populations. For instance, gene flow from the Sumatran to the Bornean population would reduce the frequency of LoF variants in Borneo, whereas gene flow from Borneo to Sumatra would introduce new LoF variants in Sumatra. It is, however, noteworthy that in the absence of fitness effect estimates associated with these LoF variants, our estimates should only be considered as approximations of individual mutational load.

Recent genomic studies indicate that attempts at genetic rescue from outbred populations can have detrimental effects by increasing mutational load of small and inbred populations[24,25], which suggest that genetic rescue attempts should be avoided in some cases. However, this claim is in conflict with a large number of studies showing that assisted gene flow most often results in a genetic rescue effect[27]. Our analyses highlight the usefulness of genomic data to estimate individual mutational load, which subsequently can be used to assess the genetic consequences of assisted gene flow[27].

In summary, our results suggest that from a conservation perspective, the current genetic status of Sumatran rhinoceros is at odds with the extremely small extant populations and associated concerns about inbreeding depression. However, the temporal changes observed in the Malay Peninsula population may serve as a warning of what might occur in the near future in the two surviving populations. Our results suggest that during the 20th century, the Malay Peninsula population experienced an increase in inbreeding, and probably also inbreeding depression followed by purging of mutational load as indicated by the comparatively low level of mutational load shortly prior to its extinction. With little evidence for recent inbreeding in the two surviving populations and with few fixed deleterious alleles, the long-term survival of the Sumatran rhinoceros does not seem to be immediately threatened by detrimental genetic factors characteristic of small populations. However, given the extremely small sizes of the surviving populations on Borneo and Sumatra, it appears inevitable that inbreeding will increase in the near future, which in turn is likely to also expose the recessive deleterious alleles in homozygous state, thus reducing population viability even further. Assisted gene flow guided by genome sequencing has the potential to help mitigate such a process.

## Methods

**Bioinformatics processing of de novo assembly.** The de novo reference genome for Sumatran rhinoceros (*D. sumatrensis harrissoni* GCA_014189135.1[71]) was used to map all re-sequencing data (see Supplementary Methods). By blasting the genome against the horse X chromosome, two sex-linked scaffolds were identified. In addition, repeats and transposable elements were predicted from the genome assembly, and the assembly was masked using the predicted repeat library as input. Finally, CpG sites (all sites where a C nucleotide is followed by a G nucleotide in the reference genome) were identified using a custom script (https://github.com/tvdvalk/find_CpG).

**Sample and library preparation for re-sequencing.** Thirty-one historical bone, skin and tooth samples from Borneo, Sumatra and the Malay Peninsula collected between 1868 and 1921 were obtained from five European natural history museums (Supplementary Table 1). Based on high endogenous DNA content (i.e., 36–89%; Supplementary Table 2), double-stranded libraries from five specimens from Borneo and the Malay Peninsula, were re-sequenced on an Illumina HiSeqX with a 2 × 150 bp setup in the High Output mode at the National Genomics Infrastructure (NGI), the sequencing facility of Science for Life Laboratories (SciLifeLab) in Stockholm, Sweden (see Supplementary Methods)[72,73]. Sixteen modern DNA extracts were sent to NGI Stockholm for library preparation and re-sequencing on an Illumina HiSeqX with a 2 × 150 bp setup in High Output mode. Utilization of samples was compliant with applicable regulatory procedures for CITES and the US Endangered Species Act. Export of blood and tissue samples for DNA extraction and genome sequencing from Sumatran rhinoceros individuals from Sabah to the Swedish Museum of Natural History, Stockholm, Sweden, was approved by The Sabah Biodiversity Council and the director of Sabah Wildlife

Department in 2014 (Licence Ref JKM/MBS.1000-2/2 (373, CITES import/export permit numbers 51138-14/0736)). Exports of DNA extracts for genome sequencing from the San Diego Zoo Global Frozen Zoo®, USA, to the Swedish Museum of Natural History, Stockholm, Sweden, was between two CITES-registered institutions (COSE transfer, CITES exemption reference number 30-3314/99), as well as under Mutual Transfer Agreement (request number BR2016005).

**Bioinformatics processing of re-sequencing data.** Adapter-trimmed historical and modern sequencing reads were mapped against the reference genome for Sumatran rhinoceros and subsequently coordinate sorted, indexed, and PCR duplicates were removed from the alignments (see Supplementary Methods).

Read group information, including library, lane and sample identity, was assigned to each bam file (see Supplementary Methods). Reads were then re-aligned around indels, and only read alignments with mapping quality ≥ 30 were kept for subsequent analysis. Three specimens (SR01, OR2142 and Gelugob) had very low coverage of 3X, 5X and 2X, respectively (Supplementary Table 2). For the remaining dataset, the average genome coverage ranged from 17X to 29X in modern and from 9X to 13X for historical genomes.

Variant calling was done in historical and modern genomes, and resulting vcf files were filtered for a minimum depth of coverage of 1/3X of the average coverage, base quality ≥ 30 and SNPs within ± 5 bp of indels were removed. CpG sites and repeat regions identified in the reference genome (see 'Bioinformatics processing of de novo assembly' section) and two X chromosome-linked scaffolds were excluded from the vcf files. All 21 rhinoceros were included for population structure analyses (i.e., PCA, ADMIXTURE), using 3,568,319 high-quality SNP calls. For all other analyses (genome-wide diversity, inbreeding, mutational load, variants in coding regions, tests of positive selection) the 18 rhinoceroses with coverage ≥ 9X were included, using 4,656,534 high-quality SNP.

### Data analysis

*Population structure.* An unrooted phylogeny was built by inferring genotype posterior probabilities for each individual, estimating pairwise genetic distances, and then estimating phylogeny from the distance matrix (see Supplementary Methods). Second, a principal component analysis (PCA)[74] was performed, and genetic clusters K ranging from 1 to 6[75] were identified.

*Demographic reconstruction and population divergence.* The pairwise sequentially Markovian coalescent[42] model was used to infer fluctuations in $N_e$ of the three major lineages of Sumatran rhinoceros over time (see Supplementary Methods). 100 bootstrap replicates were conducted per individual, using the intermediate substitution rate of $2.34 \times 10^{-8}$ substitutions/site/generation from the ones compared in Mays et al.[30] and a generation time of 12 years[76]. The split time ($T$) between each population pair of the Sumatran rhinoceros populations was estimated by assuming no coalescent events since divergence between the populations and using the PSMC approach applied to a pseudodiploid X chromosome genome.

*Genome-wide heterozygosity and runs of homozygosity (ROH).* Individual genome-wide autosomal heterozygosity was estimated using mlRho, which uses the estimated population mutation rate ($\theta$) to approximate heterozygosity (i.e., heterozygous sites per 1000 bp) under the infinite sites model[44]. Two-sided pairwise $t$-tests were used in R[77] to statistically compare $\theta$ between groups.

Two different approaches were used to identify runs of homozygosity (ROH)[78,79]. Based on these results, the individual inbreeding coefficient $F_{ROH}$ was estimated as the overall proportion of the genome contained in ROH for (1) ROH ≥ 100 kb and (2) ROH ≥ 2 Mb. To statistically compare $F_{ROH}$ between groups, two-sided pairwise $t$-tests were used in R[77].

*Mutational load based on evolutionary constrained regions.* An estimate of genome conservation across evolutionary time was used as a proxy for the deleteriousness of genomic variants. For each individual, the total number of homozygous and heterozygous derived alleles were obtained and stratified by GERP-score[47] within highly conserved regions of the genome (excluding sites with missing genotypes; see Supplementary Methods). Individual relative mutational load was measured as the sum of all derived alleles multiplied by their GERP-score, only including derived alleles above a GERP-score of 4, divided by the total number of derived alleles per individual. The percentage of derived alleles unique to each population or shared between populations at high GERP scores (> 4; i.e., those putatively deleterious), was also calculated. This was done by randomly subsampling six alleles at each genomic site with a GERP-score above 4 from each of the modern populations (thus three samples per population to exclude sample biases), and counting how often a derived allele was unique to a specific population or shared with one or both of the other populations. Two-sided pairwise $t$-tests were used in R[77] to statistically compare individual relative mutational load between groups.

*Mutational load in coding regions and missense variants.* Synonymous and non-synonymous nucleotide substitutions within coding regions as well as substitutions in proximity of coding regions were annotated for the 18 high-coverage modern and historical Sumatran rhinoceros genomes, by mapping the genomes to the white rhinoceros (*C. simum simum*) genome assembly and using its annotation of 33,026

genes (GCF_000283155.1 see Supplementary Methods)[80]. Variants were identified in three different categories: (a) Synonymous: mostly harmless or unlikely to change protein behaviour; (b) Missense: non-disruptive variants that might change protein effectiveness; and (c) LoF: variants assumed to have high (disruptive) impact on the protein, probably causing protein truncation, LoF or triggering nonsense-mediated decay (e.g., stop codons, splice donor variant and splice acceptor)[80]. The variants in these four categories were also differentiated by homozygous and heterozygous state.

Two types of comparisons were performed: (1) between the modern and historical specimens for the Bornean ($n = 5$) and the Malay Peninsula ($n = 6$) populations, and (2) among modern samples from the Bornean ($n = 4$), Malay Peninsula ($n = 3$) and Sumatran ($n = 8$) populations. The number of variants among populations was then compared using two-sided pairwise $t$-tests in R[77]. For the comparison among modern samples, the number of LoF variants shared among and unique to each population, as well as the estimated difference in frequency of LoF variants among populations were reported.

Finally, the per-individual identified LoF variants were used to predict the risk of introducing new LoF variants in a receiving population in the case of translocation of individuals. This was done by counting the number of LoF variants in each individual, then estimating how many of them were absent (allele frequency = 0) in the other two populations.

*Detecting positive selection.* First, for each population, the frequency of variants characterised as missense to identify genes potentially involved in local adaptation in modern populations was estimated and the number of variants among populations statistically compared as described in the previous section. We also reported the number of missense variants common or unique to each population.

Second, the population branch statistic (PBS) was estimated to investigate adaptation to local environments in the three Sumatran rhinoceros populations (see Supplementary Methods). Given the limited sample size, PBS values can get large due to extreme allele frequency differences. For example, if a gene's $F_{ST}$ between the target and the sister population is 1, with both populations fixed for a different allele, the PBS value can be infinite. Thus, infinite PBS values were replaced with the maximum value of the distribution of PBS values for that population. Finally, all genes with a PBS value larger than 3, ca. the 99.8th quartile of the distribution of the PBS values of all genes for each population, were reported.

*Gene Ontology analysis.* We assessed the biological functions associated with LoF and missense variants as well as for genes identified with the PBS approach and tested for statistical overrepresentation for each of these categories using horse (*Equus caballus*) as reference set (see Supplementary Methods).

**Reporting summary**. Further information on research design is available in the Nature Research Reporting Summary linked to this article.

## Data availability

The *D. sumatrensis harrissoni* assembly and the *C. simum simum* assembly are available at NCBI (*D. sumatrensis harrissoni*: BioProjectID: PRJNA638009, assembly: GCA_014189135.1, *C. simum simum*:, BioProjectID: PRJNA74583, assembly: GCF_000283155.1). Modern and historical re-sequencing data (fastq files) generated for this project are available at the European Nucleotide Archive (project accession number PRJEB35511). Source data are provided with this paper.

## Code availability

The three Python scripts used in this study and are available on GitHub: https://github.com/tvdvalk/find_CpG[4], https://github.com/pontussk/samremovedup[81], and https://github.com/mathii/gdc.git.

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

## Acknowledgements

L.D. and J.vS. acknowledge support from FORMAS (grant 2015-676). N.D. was funded by the Swiss National Science Foundation (Early Postdoc Mobility grant P2SKP3_165031 and Advanced Postdoc Mobility grant P300PA_177845) and the Carl Tryggers Foundation (CTS 19:257). D.D.dM. was supported through a Carl Tryggers scholarship (grant CTS17:109). K.G. acknowledges support from FORMAS (grant 2016-00835). P.S. acknowledges the European Research Council (grant 852558), the Wellcome Trust (217223/Z/19/Z) and Francis Crick Institute core funding (FC001595). M.T.P.G. acknowledges ERC Consolidator Grant 681396 'Extinction Genomics'. Y.C. acknowledges support from the EU Marie Curie Action (FP7-MC-IIF project 301572). V.K. and M.K. are financially supported by the Knut and Alice Wallenberg Foundation as part of the National Bioinformatics Infrastructure Sweden at Science for Life Laboratories. C.W.W. acknowledges the Swedish Research Council (2017-04386). We thank Roberto Portela-Miguez, Tom Geerinckx, Daniela Kalthoff, Frank Zachos, Alexander Bibl, and Eline Lorenzen for assisting with sampling of museum specimens from the NHM London, RBINS Brussels, NRM Stockholm, NMW Vienna, and NHM Denmark, respectively. We also thank Zainal Zainuddin and John Payne from Borneo Rhino Alliance (BORA) and the late Diana Angeles Ramirez Saldivar from the Wildlife Rescue Unit for providing modern samples. We thank the Sabah Biodiversity Centre and the Sabah Wildlife Department for allowing us to export samples from Sumatran rhinoceros individuals from Sabah. We thank the Zoological Society of San Diego and San Diego Zoo Global Frozen Zoo® for allowing us to utilize frozen samples from Sumatran rhinoceros individuals from Malay Peninsula, Sumatra, and Borneo. Sequencing was performed by the Swedish National Genomics Infrastructure (NGI) at the Science for Life Laboratory, which is supported by the Swedish Research Council and the Knut and Alice Wallenberg Foundation. The data handling was enabled by resources provided by the Swedish National Infrastructure for Computing (SNIC) at the Uppsala Multidisciplinary Centre for Advanced Computational Science (UPPMAX) partially funded by the Swedish Research Council through grant agreement no. 2018-05973, under projects SNIC 2020/5-3 and 2019/8-331. Finally, we thank Eleftheria Palkopoulou for providing advice on bioinformatics analyses.

## Author contributions

J.vS., L.D., Y.C. and B.G. conceived the study, with N.D. and D.D.dM. contributing advice on the study design. S.K.S.S.N., B.G., S.B., O.R., M.T.P.G., L.D. and A.G. contributed with samples, important materials and/or resources. J.vS., L.D., C.S. and M.-H.S.S. performed DNA extractions and library preparation. J.vs., N.D., D.D.dM., T.vV., V.E.K, M.K. and S.L. performed bioinformatics and computational analyses of the data. L.D., C.W.W., K.G., P.S. and S.P. provided advice on the data analyses. J.vS. and N.D. drafted the manuscript. All authors contributed to the final version of the manuscript.

## Competing interests

The authors declare no competing interests.
