## [Peer Review File · Nature Communications]

Reviewers' Comments:

Reviewer #1:

Remarks to the Author:

I spent a long (too long) time looking for what should have been obvious almost immediately. "What was the genomic insight" from the project? Perhaps it is the different patterns of inbreeding and mutational load?

The manuscript presents a thorough analysis of WGS sequence data from 3 populations of Sumatran rhinoceros. It is a well-written summary of the process of WGS analyses and results but remarkably dry/void of basic biology / bio geography (although for the most part there are citations one can go to for the information.

The abstract is so abstract to be almost uninformative. Numbers of individuals, from what material, what sort of sequence data, descriptors such as "small", "little", "low", "genomic consequences" provide little usable information.

This writing style continues through the Introduction, e.g. "vulnerable to several extrinsic and intrinsic threats such as environmental effects" provides no information. The first 5-6 paragraphs would be more useful if they provided more specific taxonomic, population, and biogeographic context. Some of this information is in the discussion, but some readers will not get that far.

More information on the samples used, with specifics for each, should be clearly presented so that results can be easily compared based on tissue types used. Many readers will be very interested in the approaches used to collect and compare these different samples.

In the Discussion (especially 2 and 3rd paragraphs), check the use and distinction of when to use "between" and "among".

Perhaps the best way forward is to present paper and results as a biogeographical question with conservation implications, and then present more information on in situ and ex situ status and implications. As such it would also attract a broader readership.

Reviewer #2:

Remarks to the Author:

This manuscript describes genome sequencing of historical and modern endangered Sumatran rhinoceroses from Borneo, Malaysia (extinct) and Sumatra. From the sequences, they estimated genomic diversity, inbreeding (FROH) and mutational loads.

This is an interesting case study with strong conservation implications that will appeal to a wide audience. Their conclusion that gene flow between Bornean and Sumatran rhinoceroses is desirable is justified (lines 388-403).

As best I can tell the genomics is very well done. The manuscript is mostly well written and suitable for the journal's audience.

However, I have queries about the treatments of deleterious alleles. To assess their impacts, we need to know the effects, the frequencies of alleles and the number of loci involved. Numbers of fixed deleterious alleles in the three population are provided, but the distributions of frequencies of deleterious alleles at the polymorphic loci are not presented and this means the information is incomplete.

Detailed comments:

Line 70: It is not only the large-effect deleterious alleles that are of conservation concern, but the larger number of deleterious alleles with lesser effects are also of major concern as potential causes of

inbreeding depression. There are populations with few large-effects deleterious harmful alleles that still show substantial inbreeding depression.

Line 77: the authors use "translocation", but that term encompasses movement to a location without any animals of the species. Their meaning is "translocation and gene flow" here and elsewhere. I suggest they use "gene flow" throughout.

Lines 82-84: "hybrid vigor is mostly considered not to persist beyond a few generations". This is misleading. Theory shows that benefits are expected to persist beyond the F3 generation in outbreeding species (as here) provided population sizes are large (Frankham 2015). Further, there is empirical data on persistence of the benefits of gene flow for generations beyond the F3, based on the meta-analysis of Frankham (2016): this shows persistence that extends to F16 and the effects are significantly beneficial (Frankham et al. 2019, p. 72).

Line 93: The number of genetic rescues for conservation purposes has increased significantly since 2015 and is now ~34 (Frankham et al. 2019 p. 8-9).

Lines 96-101: the conservation implication of Kyriazis et al. (2019 unreviewed pre-print) and Robinson et al. (2019) (especially the simulations) are scientifically unsound. A critique paper has just appeared in *Biological Conservation* by Ralls et al. (2020) pointing out problems with their work.

Line 132: "incentives" – this has an inappropriate meaning. I suggest that the authors replace it with "proposals" (or "suggestions")

Lines 204-241: Mutational load: The treatment of mutational load is hard to follow as there are three aspect to the load, allele frequencies, allele effects, and number of loci with deleterious alleles that are needed to predict the impact of harmful alleles on fitness. The treatment of homozygous alleles is clear. However, I struggled to understand what number of deleterious alleles meant (a common problem with genomic papers), as the same numbers can be produced by one harmful allele per locus, or 10 per locus at one tenth the number of loci, etc. From the information on lines 267-272, there are an average of 2.4 copies of missense variants over 6,490 loci, so it is not one copy per locus. I recommend adding a figure indicating the distribution of frequencies of harmful alleles across loci.

Lines 242-247: Translocation of a single individual is hardly adequate for conservation purposes unless it is done persistently over time.

Lines 272-247 & 351-389: Impact of gene flow: In assessing the impact of gene flow it is critical that both the potential costs and the potential benefits are considered. This manuscript mainly considers the potential costs (the new harmful alleles introduced). Thus, there needs to be an assessment of the simultaneous reduction in frequency of pre-existing harmful alleles in the recipient alleles i.e. the introduction of beneficial alleles. For example, 382 fixed harmful missense alleles in the Bornean populations will become polymorphic on crossing to the Sumatran population, substantially reducing their fitness impacts. Further, other harmful alleles are likely to have their frequencies substantially reduced. On the harmful front, the ms indicates that on average 10 loss of function alleles would be added by crossing, but 7 Bornean loci would go from homozygous to polymorphic on crossing to the Sumatran populations. The cost and benefits are different for gene flow from the Bornean population into the Sumatran one.

References

Frankham, R., 2016. Genetic rescue benefits persist to at least the F3 generation, based on a meta-analysis. *Biological Conservation* 195, 33-36.

Frankham, R., Ballou, J.D., Ralls, K., et al., 2019. *A Practical Guide for Genetic Management of Fragmented Animal and Plant Populations*. Oxford University Press, Oxford, UK.

Ralls, K., Sunnucks, P., Lacy, R.C., et al., 2020. Genetic rescue: A critique of the evidence supports maximizing genetic diversity rather than minimizing the introduction of putatively harmful genetic variation. *Biological Conservation* 251, 108784.

Reviewer #3:

Remarks to the Author:

Von Seth et al. 2020 Genomic insights into the conservation status of the world's last remaining

Sumatran rhinoceros populations. Nature Communications.

I see no glaring omissions or errors in the methods but admittedly I am relatively new to genome-level analyses. Everything in terms of the methodology seems done in a thorough and straightforward manner using approaches like PSMC, Structure, GERP, etc that are well trodden paths in this field. Overall this is an excellent contribution to our knowledge of *Dicerorhinus* in Sundaland.

Lines 56-57. I would maybe have the latter part of the last sentence read "...could potentially be mitigated by outbreeding among populations."

Lines 131-139. I don't think the taxonomic implications should be downplayed. There to date have been several genetic studies on Sumatran Rhinoceros including studies employing mtDNA and microsatellites showing fairly distinct genetic lineages of Sumatran Rhinoceros between Sumatra, Borneo, the Malay Peninsula (see Steiner et al. 2017, Brandt et al. 2018, Morales et al. 1997, and even some suggestive data in the current manuscript) and fossil evidence of a more diverse *Dicerorhinus* in Mainland Asia (*D. gwebinensis* in Myanmar and *D. lantianensis* and *D. yunchuchenensis* in China, see Tong 2012, Antoine 2012). Given these genetic data, past species diversity within *Dicerorhinus* and the clear and often deep species divergences among other codistributed vertebrates spanning Indochina, the Malay region and the Sunda shelf it's likely the current allopatric subspecies of Sumatran rhinoceros warrant elevation to species status but to date no one has done a rigorous test of species limits in Sumatran Rhinoceros despite the obvious important conservation implications and the recent availability of multiple genome wide datasets. Model testing approaches that employ genome wide datasets and can account for speciation with gene flow like those available in PHRAPL (Jackson et al. 2017) would be very useful in testing whether or not these Sumatran Rhinoceros populations represent one, two or three historical species lineages. This question of species limits in Sumatran rhinoceros is I think at the heart of the ultimate conservation and management question. Are we willing to essentially erase two evolutionarily independent lineages through gene flow in a management effort to preserve something along the lines of a Sumatran rhinoceros? I believe this is a cost worth paying but we must go into those efforts clear-eyed as to what we are doing and first and foremost that requires understanding if these populations represent different species lineages (with species being something along the lines of species as evolutionarily independent lineages ala de Queiroz et al. 2007). I believe that the current paper takes the subspecies ranks (*sumatraensis*, *harrisoni*, and the unsampled and almost certainly extinct *lasiotis*, the latter of which is left unmentioned in this paper) a bit for granted and not as largely untested hypotheses.

Lines 182-191. Some additional discussion of correlates of these split times may be merited. The split between Sumatra/Malay and Borneo populations at 300k years BP would I believe fell in a warmer interglacial period but not at a period where the Sunda shelf was completely submerged meaning that perhaps other isolating barriers existed between Borneo and Sumatra/Malay Peninsula populations, such as an unsuitable Sunda savannah corridor. This someone porous barrier during a time during the Pleistocene where the Sunda shelf was not completely submerged but terrestrial with savannah habitat reticulated by riparian forest corridors may account for the finding that isolation between Borneo and Sumatra/Malay was not abrupt but rather gradual.

Lines 197-203. Would it be too much to include the per 1kb heterozygosity estimates for each population in the text rather than the relative differences (or in addition to the relative differences) without having the reader go to supplementary table 6? I would note that the genome-wide heterozygosity values are roughly similar to the genome sequence from the *D. s. sumatrensis* genome from Sumatra described in Mays et al. 2018 (1.3 per 1kb).

Lines 204-233. Maybe some discussion of this idea of mutational load and LoF in this context is warranted? This discussion would be relevant for lines 351-373. The authors are attributing these differences in load and LoF to either neutral or deleterious demographic processes like inbreeding or drift but LoF itself may be considered adaptive. Selection may favor the loss of an adaptive life history

character such as migration when a species colonizes a more seasonally stable environment leading to a relaxation of selection on a suite of traits associated with migration. Likewise relaxed selection related to changing epistatic interactions among traits and energetic trade-offs related to the loss of eyes and pigments in cave species relative to their ancestral surface populations would lead to many LoF genes associated with these traits but this would be the result of selection. I suspect that it may not be so straightforward to say that these LoF genes or the PBS approach may not detect all kinds of variation resulting from selection. Sumatran Rhinoceros is an interesting case being derived from a clade of grazing temperate and arctic ancestors (other closely related dicerorhine rhinoceroses such as *Coelodonta* and *Stephanorhinus* were more temperate in their distribution and relied more, or even entirely, on grasses) and evolving to adapt to become a tropical forest browser and this adaptive trajectory may have promoted the loss of genes related to living in those more temperate environments where grazing was a bigger part of the diet. I would be curious to know how the Venn diagram in Supplementary figure 15 compares to other sister taxa of mammals (maybe sister taxa with the ungulates). Is what is displayed here more or less than we see for other allopatric and similarly divergent mammals? I know there is a very thorough study comparing genomes among *Panthera* species by Figueiró et al. 2017 but I don't think they explored variation in LoF? It is interesting that the lowest number of LoF variants is associated with the population, Malay, with the highest inbreeding coefficient and the lowest population scaled mutation rate (θ) while the greatest number of LoF variants is associated with the population with the lowest inbreeding coefficient and comparatively high θ (see lines 234-241 and supplementary figures 7-10). This would seem the opposite of what one would expect if these LoF variants are the result of neutral demographic processes or the result of inbreeding?

Lines 274-284. There is some justification based on small sample size regarding these fixed differences among populations for these 33,026 genes however I would maybe exercise some caution in both directions. These findings indeed could hint at these populations (particularly Borneo vs Sumatra/Malay) being separate lineages with a unique and evolutionarily divergent and independent history but alone these results are not compelling one way or the other. These results just bring more attention to the need for a more rigorous species delimitation analysis.

Lines 300-310. I would note that population structure may be difficult to distinguish from a bottlenecked or declining population in a PSMC analysis (see Gautier et al. 2016, Nadachowska-Brzyska et al. 2016). There is also no small amount of debate around the species limits proposed in the Sumatran orangutans, paraphyly and the distinction between population structure and species limits.

I think a fundamental question here regards the management decision whether or not to deliberately introgress genes between these populations. The authors seem to be only addressing the question of whether it is reasonable to conclude that this introgression would not result in any significant decrease in viability, which, given the Sumatran rhinoceros' precarious situation, may arguably be the only relevant question. However, the question that is left unanswered is whether it is warranted that these two populations should be considered separate species lineages (whether *lasiotis* is a species lineage or not is an interesting question but not one that is relevant in this study). That question has not been addressed. I think the management plan for Sumatran Rhinoceros should be clear as to what we are doing. Would translocating individuals among breeding populations be merging two evolutionarily independent population lineages (i.e. species ala de Queiroz 2007)? I believe whether the decision to translocate breeding individuals among populations would affect viability as an absolutely critical but separate question. If we may reasonably demonstrate that translocation will have no deleterious effects in terms of population viability and translocations are the only way in which any Dicerorhine rhinoceros will survive then that may override the effect of losing two endemic species (one in Borneo and one in Sumatra/Malay) to a new hybrid lineage, but it would be nice up front to know if that is indeed what we are doing. Maybe a rigorous analysis of this data specifically focusing on species limits using Bayesian and other model-based approaches is beyond the scope of this paper and a topic for another submission but the fundamental issue remains as to whether management decisions to adopt

a genetic rescue plan and introgress the two populations would in effect be leading to the genetic extinction of two distinct species so that the genus may survive. I don't think that plan is unreasonable under the circumstances, but we should enter into that plan knowing what we are doing.

REVIEWER COMMENTS

Reviewer #1:

Reviewer #1 comment: I spent a long (too long) time looking for what should have been obvious almost immediately. "What was the genomic insight" from the project? Perhaps it is the different patterns of inbreeding and mutational load?

The manuscript presents a thorough analysis of WGS sequence data from 3 populations of Sumatran rhinoceros. It is a well-written summary of the process of WGS analyses and results but remarkably dry/void of basic biology / bio geography (although for the most part there are citations one can go to for the information).

The abstract is so abstract to be almost uninformative. Numbers of individuals, from what material, what sort of sequence data, descriptors such as "small", "little", "low", "genomic consequences" provide little usable information.

This writing style continues through the Introduction, e.g. "vulnerable to several extrinsic and intrinsic threats such as environmental effects" provides no information. The first 5-6 paragraphs would be more useful if they provided more specific taxonomic, population, and biogeographic context. Some of this information is in the discussion, but some readers will not get that far.

More information on the samples used, with specifics for each, should be clearly presented so that results can be easily compared based on tissue types used. Many readers will be very interested in the approaches used to collect and compare these different samples.

Response: *We thank the reviewer for pointing out that the goal of the paper is not clear, and that it would benefit from more detailed information. We have now clarified the overall goal (to use both historical and modern genomes to infer conservation-related consequences of a recent decline in Sumatran rhinoceros) in the Abstract on l. 50-51, have added the number of genomes sequences on l. 49, and the type of tissues used in Supplemental Table 1. We have also added more information relevant to the biogeography of the species in the introduction on l. 122-126,149-154.*

In the Discussion (especially 2 and 3rd paragraphs), check the use and distinction of when to use "between" and "among".

Response: *We have now corrected the use of 'between' and 'among' throughout the manuscript.*

Perhaps the best way forward is to present paper and results as a biogeographical question with conservation implications, and then present more information on in situ and ex situ status and implications. As such it would also attract a broader readership.

Response: *While we agree with the reviewer that the taxonomic and biogeographical context is important, we want to keep the focus on conservation-related genomic parameters. We have, however, added more information on the biogeography of the species in the introduction on l. 122-126,149-154. We also discuss the taxonomic implications for *D. sumatrensis* on l. 354-366.*

#####

Reviewer #2:

This manuscript describes genome sequencing of historical and modern endangered Sumatran rhinoceroses from Borneo, Malaysia (extinct) and Sumatra. From the sequences, they estimated genomic diversity, inbreeding (FROH) and mutational loads.

This is an interesting case study with strong conservation implications that will appeal to a wide audience. Their conclusion that gene flow between Bornean and Sumatran rhinoceroses is desirable is justified (lines 388-403).

As best I can tell the genomics is very well done. The manuscript is mostly well written and suitable for the journal's audience.

However, I have queries about the treatments of deleterious alleles. To assess their impacts, we need to know the effects, the frequencies of alleles and the number of loci involved.

Numbers of fixed deleterious alleles in the three population are provided, but the distributions of frequencies of deleterious alleles at the polymorphic loci are not presented and this means the information is incomplete.

Response: *The reviewer is correct that the impact of deleterious variants is difficult to predict, especially without phenotype data. However, mutational load (i.e. number of deleterious variants) can be used as a proxy for the fitness of a population since it only refers to the accumulation of deleterious mutations. We now make a distinction between genetic load and mutational load in the introduction on l. 112-116 and also re-iterate in the result section why we are focusing on estimating mutational load on l. 228-230. We understand that the reviewer wants us to provide the reader with the frequency distributions of LoF and missense variants (i.e. SNP). The allele frequencies for all LoF or missense variants are provided in Supplemental Table 6, with the information about the genes containing these variants. If we have misunderstood what the reviewer is asking for, we kindly ask for a clarification and will reconsider the comment.*

Detailed comments:

Line 70: It is not only the large-effect deleterious alleles that are of conservation concern, but the larger number of deleterious alleles with lesser effects are also of major concern as

potential causes of inbreeding depression. There are populations with few large-effects deleterious harmful alleles that still show substantial inbreeding depression.

Response: *The reviewer is correct and we have now edited this statement accordingly on l. 70-72.*

Line 77: the authors use “translocation”, but that term encompasses movement to a location without any animals of the species. Their meaning is “translocation and gene flow” here and elsewhere. I suggest they use “gene flow” throughout.

Response: *We have now replaced ‘translocations’ with ‘assisted gene flow’ where appropriate as suggested by the reviewer.*

Lines 82-84: “hybrid vigor is mostly considered not to persist beyond a few generations”. This is misleading. Theory shows that benefits are expected to persist beyond the F3 generation in outbreeding species (as here) provided population sizes are large (Frankham 2015). Further, there is empirical data on persistence of the benefits of gene flow for generations beyond the F3, based on the meta-analysis of Frankham (2016): this shows persistence that extends to F16 and the effects are significantly beneficial (Frankham et al. 2019, p. 72).

Response: *we thank the reviewer for pointing this out. We have rephrased this statement accordingly on l. 85-86.*

Line 93: The number of genetic rescues for conservation purposes has increased significantly since 2015 and is now ~34 (Frankham et al. 2019 p. 8-9).

Response: *We have now updated this number and the reference on l. 95.*

Lines 96-101: the conservation implication of Kyriazis et al. (2019 unreviewed pre-print) and Robinson et al. (2019) (especially the simulations) are scientifically unsound. A critique paper has just appeared in Biological Conservation by Ralls et al. (2020) pointing out problems with their work.

Response: *We now discuss the criticism by Ralls et al. (2020). We have added in the introduction and in the discussion that e.g. ‘It is thus essential to weigh the positive and negative effects of this alternative approach when assessing the need for genetic rescue’ on l. 105-109 and 463-466, respectively.*

Line 132: “incentives” – this has an inappropriate meaning. I suggest that the authors replace it with “proposals” (or “suggestions”)

Response: *We have replaced ‘incentives’ with ‘proposals’ as suggested by the reviewer on l. 147.*

Lines 204-241: Mutational load: The treatment of mutational load is hard to follow as there are three aspects to the load, allele frequencies, allele effects, and number of loci with deleterious alleles that are needed to predict the impact of harmful alleles on fitness. The treatment of homozygous alleles is clear. However, I struggled to understand what number of deleterious alleles meant (a common problem with genomic papers), as the same numbers can be produced by one harmful allele per locus, or 10 per locus at one tenth the number of loci, etc. From the information on lines 267-272, there are an average of 2.4 copies of missense variants over 6,490 loci, so it is not one copy per locus. I recommend adding a figure indicating the distribution of frequencies of harmful alleles across loci.

Response: *Here, we are under the impression that the reviewer is referring to genetic load (i.e. the difference between the fitness of an average genotype in the population and the fitness of a reference genotype). Without any information on the effect of deleterious mutations on fitness, we cannot estimate genetic load. However, we can count the number of deleterious variants (i.e. SNPs) and quantify changes in mutational load to investigate the effects of population decline in Sumatran rhinoceros. We now make the distinction between genetic load (i.e. the decrease in fitness of the average individual in a population relative to the fittest genotype due to the presence of deleterious genes in the gene pool) and mutational load (i.e. number of deleterious variants/SNP) in the introduction on l. 66-67 and 114. We also have clarified in the introduction and in the result section why we focus here on mutational load on l. 112-116 and l. 228-230. Finally, the reviewer is right that the distribution of the number of deleterious alleles per gene can differ and can have an effect on the impact of mutational load, and is thus relevant when analyzing differences in mutational load between the populations. We thank the reviewer for pointing this out, and we have now added a supplementary table (Supplementary Table 8) showing the number of genes with 1 to 6 LoF variants (the maximum number of LoF variants per gene was 6) in each population. The vast majority of genes carrying LoF variants had only one variant per gene. The Sumatran population had one gene with 6 LoF variants. The three populations had roughly the same number of genes with 3 LoF variants, and Borneo (the population with the highest total number of LoF variants) had the highest number of genes with 1 and 2 LoF variants per gene while Malay Peninsula (the population with the lowest total number of LoF variants) had the lowest. We've commented on the new table on l. 254-257, stating that most genes had one LoF variant per gene and that the highest number of LoF variants per gene was six.*

Lines 242-247: Translocation of a single individual is hardly adequate for conservation purposes unless it is done persistently over time.

Response: *The reviewer is correct. Here, we are not trying to say that translocating one individual would result in genetic rescue. However, since the number of individuals surviving in the wild is very small and since obtaining gametes for large*

number of individual is unlikely, it would be informative for conservation biologist in charge of future translocations programs to know which individuals are more likely to lead to an increase in hybrid vigor instead of an increase in mutational load. We have now added a sentence to clarify the purpose of this analysis on l. 272-276.

Lines 272-247 & 351-389: Impact of gene flow: In assessing the impact of gene flow it is critical that both the potential costs and the potential benefits are considered. This manuscript mainly considers the potential costs (the new harmful alleles introduced). Thus, there needs to be an assessment of the simultaneous reduction in frequency of pre-existing harmful alleles in the recipient alleles i.e. the introduction of beneficial alleles. For example, 382 fixed harmful missense alleles in the Bornean populations will become polymorphic on crossing to the Sumatran population, substantially reducing their fitness impacts. Further, other harmful alleles are likely to have their frequencies substantially reduced. On the harmful front, the ms indicates that on average 10 loss of function alleles would be added by crossing, but 7 Bornean loci would go from homozygous to polymorphic on crossing to the Sumatran populations. The cost and benefits are different for gene flow from the Bornean population into the Sumatran one.

Response: *We have now clarified these points and highlighted how detrimental homozygous variants can be reduced in frequency through gene flow on l.454-457. We also reiterate that the risks and benefits of alternative approaches to gene flow should be weighed when considering the need for gene flow by referring to Ralls et al. (2020) on l. 105-109 and 463-466.*

References

- Frankham, R., 2016. Genetic rescue benefits persist to at least the F3 generation, based on a meta-analysis. *Biological Conservation* 195, 33-36.
- Frankham, R., Ballou, J.D., Ralls, K., et al., 2019. *A Practical Guide for Genetic Management of Fragmented Animal and Plant Populations*. Oxford University Press, Oxford, UK.
- Ralls, K., Sunnucks, P., Lacy, R.C., et al., 2020. Genetic rescue: A critique of the evidence supports maximizing genetic diversity rather than minimizing the introduction of putatively harmful genetic variation. *Biological Conservation* 251, 108784.

#####

Reviewer #3:

Von Seth et al. 2020 Genomic insights into the conservation status of the world's last remaining Sumatran rhinoceros populations. *Nature Communications*.

I see no glaring omissions or errors in the methods but admittedly I am relatively new to genome-level analyses. Everything in terms of the methodology seems done in a thorough and straightforward manner using approaches like PSMC, Structure, GERP, etc that are well trodden paths in this field. Overall this is an excellent contribution to our knowledge of Dicerorhinus in Sundaland.

Lines 56-57. I would maybe have the latter part of the last sentence read "...could potentially be mitigated by outbreeding among populations."

Response: *we have now rephrased the sentence and now state that 'Moreover, we find little evidence for differences in local adaptation among the populations, suggesting that future inbreeding depression could potentially be mitigated by assisted gene flow among populations' on l. 57.*

Lines 131-139. I don't think the taxonomic implications should be downplayed. There to date have been several genetic studies on Sumatran Rhinoceros including studies employing mtDNA and microsatellites showing fairly distinct genetic lineages of Sumatran Rhinoceros between Sumatra, Borneo, the Malay Peninsula (see Steiner et al. 2017, Brandt et al. 2018, Morales et al. 1997, and even some suggestive data in the current manuscript) and fossil evidence of a more diverse Dicerorhinus in Mainland Asia (*D. gwebinensis* in Myanmar and *D. lantianensis* and *D. yunchuchenensis* in China, see Tong 2012, Antoine 2012). Given these genetic data, past species diversity within Dicerorhinus and the clear and often deep species divergences among other codistributed vertebrates spanning Indochina, the Malay region and the Sunda shelf it's likely the current allopatric subspecies of Sumatran rhinoceros warrant elevation to species status but to date no one has done a rigorous test of species limits in Sumatran Rhinoceros despite the obvious important conservation implications and the recent availability of multiple genome wide datasets. Model testing approaches that employ genome wide datasets and can account for speciation with gene flow like those available in PHRAPL (Jackson et al. 2017) would be very useful in testing whether or not these Sumatran Rhinoceros populations represent one, two or three historical species lineages. This question of species limits in Sumatran rhinoceros is I think at the heart of the ultimate conservation and management question. Are we willing to essentially erase two evolutionarily independent lineages through gene flow in a management effort to preserve something along the lines of a Sumatran rhinoceros? I believe this is a cost worth paying but we must go into those efforts clear-eyed as to what we are doing and first and foremost that requires understanding if these populations represent different species lineages (with species being something along the lines of species as evolutionarily independent lineages ala de Queiroz et al. 2007). I believe that the current paper takes the subspecies ranks (*sumatraensis*, *harrisoni*, and the unsampled and almost certainly extinct *lasiotis*, the latter of which is left unmentioned in this paper) a bit for granted and not as largely untested hypotheses.

Response: *We thank the Reviewer #3 for this comment. We agree that addressing the taxonomy of Sumatran rhinoceros is interesting and important for conservation, but is however not the main focus of this manuscript. It is our view that in order to resolve*

*the taxonomy properly, one would need genomes from additional extinct populations (such as Burma and China). We are in the process of accessing such samples, with the aim of resolving the Sumatran rhinoceros taxonomy for a second study. For that reason, we do not wish to go too much into depth on this matter in the current manuscript. However, since the readers are likely to be interested in the taxonomic implications of the current dataset, we have now added a brief discussion about the relatedness among the three populations analysed in this manuscript, draw a parallel with a similar situation in white rhinoceros (*Ceratotherium simum*), and also relate it to the consequences of genetic rescue on l. 354-366 and 434-444.*

Lines 182-191. Some additional discussion of correlates of these split times may be merited. The split between Sumatra/Malay and Borneo populations at 300k years BP would I believe fell in a warmer interglacial period but not at a period where the Sunda shelf was completely submerged meaning that perhaps other isolating barriers existed between Borneo and Sumatra/Malay Peninsula populations, such as an unsuitable Sunda savannah corridor. This someone porous barrier during a time during the Pleistocene where the Sunda shelf was not completely submerged but terrestrial with savannah habitat reticulated by riparian forest corridors may account for the finding that isolation between Borneo and Sumatra/Malay was not abrupt but rather gradual.

Response: *We have taken the reviewer's comment in consideration and now discuss this alternative explanation for the gradual isolation in the discussion on l. 335-339.*

Lines 197-203. Would it be too much to include the per 1kb heterozygosity estimates for each population in the text rather than the relative differences (or in addition to the relative differences) without having the reader go to supplementary table 6? I would note that the genome-wide heterozygosity values are roughly similar to the genome sequence from the *D. s. sumatrensis* genome from Sumatra described in Mays et al. 2018 (1.3 per 1kb).

Response: *We have now added these values in the text on l. 220 and 222.*

Lines 204-233. Maybe some discussion of this idea of mutational load and LoF in this context is warranted? This discussion would be relevant for lines 351-373. The authors are attributing these differences in load and LoF to either neutral or deleterious demographic processes like inbreeding or drift but LoF itself may be considered adaptive. Selection may favor the loss of an adaptive life history character such as migration when a species colonizes a more seasonally stable environment leading to a relaxation of selection on a suite of traits associated with migration. Likewise relaxed selection related to changing epistatic interactions among traits and energetic trade-offs related to the loss of eyes and pigments in cave species relative to their ancestral surface populations would lead to many LoF genes associated with these traits but this would be the result of selection. I suspect that it may not be so straightforward to say that these LoF genes or the PBS approach may not detect all kinds of variation resulting from selection. Sumatran Rhinoceros is an interesting case being derived from a clade of grazing temperate and arctic ancestors (other closely related

dicerorhine rhinoceroses such as *Coelodonta* and *Stephanorhinus* were more temperate in their distribution and relied more, or even entirely, on grasses) and evolving to adapt to become a tropical forest browser and this adaptive trajectory may have promoted the loss of genes related to living in those more temperate environments where grazing was a bigger part of the diet. I would be curious to know how the Venn diagram in Supplementary figure 15 compares to other sister taxa of mammals (maybe sister taxa with the ungulates). Is what is displayed here more or less than we see for other allopatric and similarly divergent mammals? I know there is a very thorough study comparing genomes among *Panthera* species by Figueiró et al. 2017 but I don't think they explored variation in LoF? It is interesting that the lowest number of LoF variants is associated with the population, Malay, with the highest inbreeding coefficient and the lowest population scaled mutation rate (θ) while the greatest number of LoF variants is associated with the population with the lowest inbreeding coefficient and comparatively high θ (see lines 234-241 and supplementary figures 7-10). This would seem the opposite of what one would expect if these LoF variants are the result of neutral demographic processes or the result of inbreeding?

Response: *We agree with the reviewer that LoF variants can sometimes be adaptive, as shown in (Lynch et al 2015). However, since we have excluded LoF present and fixed in all three populations, which most likely would be indicative of adaptive evolution along the whole *D. sumatrensis* lineage, we think that it is highly unlikely that the LoF variants identified in this paper have some adaptive function. From our standpoint, it seems more likely that purging has removed LoF variants from the Malay Peninsula population, rather than there being differences in the selection pressure among the populations extreme enough to make LoF variants adaptive.*

Lynch et al. (2015) Elephantid Genomes Reveal the Molecular Bases of Woolly Mammoth Adaptations to the Arctic. Cell reports 12.2 (2015): 217-228.

Lines 274-284. There is some justification based on small sample size regarding these fixed differences among populations for these 33,026 genes however I would maybe exercise some caution in both directions. These findings indeed could hint at these populations (particularly Borneo vs Sumatra/Malay) being separate lineages with a unique and evolutionarily divergent and independent history but alone these results are not compelling one way or the other. These results just bring more attention to the need for a more rigorous species delimitation analysis.

Response: *We thank the reviewer for pointing this us, and have now changed the text to make a more balanced interpretation of the results on 1316-319.*

Lines 300-310. I would note that population structure may be difficult to distinguish from a bottlenecked or declining population in a PSMC analysis (see Gautier et al. 2016, Nadachowska-Brzyska et al. 2016). There is also no small amount of debate around the species limits proposed in the Sumatran orangutans, paraphyly and the distinction between population structure and species limits.

Response: *We have now added this caveat and clarified the following sentence accordingly on l. 342-343.*

I think a fundamental question here regards the management decision whether or not to deliberately introgress genes between these populations. The authors seem to be only addressing the question of whether it is reasonable to conclude that this introgression would not result in any significant decrease in viability, which, given the Sumatran rhinoceros' precarious situation, may arguably be the only relevant question. However, the question that is left unanswered is whether it is warranted that these two populations should be considered separate species lineages (whether lasiotis is a species lineage or not is an interesting question but not one that is relevant in this study). That question has not been addressed. I think the management plan for Sumatran Rhinoceros should be clear as to what we are doing. Would translocating individuals among breeding populations be merging two evolutionarily independent population lineages (i.e. species ala de Queiroz 2007)? I believe whether the decision to translocate breeding individuals among populations would affect viability as an absolutely critical but separate question. If we may reasonably demonstrate that translocation will have no deleterious effects in terms of population viability and translocations are the only way in which any Dicerorhine rhinoceros will survive then that may override the effect of losing two endemic species (one in Borneo and one in Sumatra/Malay) to a new hybrid lineage, but it would be nice up front to know if that is indeed what we are doing. Maybe a rigorous analysis of this data specifically focusing on species limits using Bayesian and other model-based approaches is beyond the scope of this paper and a topic for another submission but the fundamental issue remains as to whether management decisions to adopt a genetic rescue plan and introgress the two populations would in effect be leading to the genetic extinction of two distinct species so that the genus may survive. I don't think that plan is unreasonable under the circumstances, but we should enter into that plan knowing what we are doing.

Response: *We now discuss and acknowledge evolutionary distinct lineages to some extent on l. 434-444, and write that the high risk for extinction of the species might warrant the risk of losing distinct evolutionary lineages.*

Reviewers' Comments:

Reviewer #1:

None

Reviewer #2:

Remarks to the Author:

Re-Review of Von Seth for Nature Communications

The authors have addressed my comments in a careful manner. In particular, the information on mutation load is much clearer and easier to follow.

However, I do have a few remaining comments as follows:

p.8 mid: In total, we found 373 LoF variants across all three populations, and found that Borneo has a significantly higher number of LoF variants compared to the other two modern populations (t-test, $p = 2.78e-05$), and that Sumatra has significantly higher number of LoF variants compared to Malay Peninsula (t-test, $p = 0.0019$,

Whilst this is true, it is misleading as the text gives values of 219, 208 and 136 for B, S, and M populations. Thus, B & S \gg M, but B is slightly $>$ S. I suggest that re-wording is justified.

p.9 Temporal Changes... and elsewhere: "heterozygosity (θ , heterozygous sites/kb)" – this form of heterozygosity is now defined in many places (I believe in response to my previous comment): It only needs to be defined once if all cases of heterozygosity are defined in the same way (which I think is the case).

p.9 bottom-10 top: "Among these, there were 1, 382, and 762 genes that had fixed variants in the Sumatra, Borneo, and Malay Peninsula populations, respectively (Supplementary Table 6)." There are two numbers and three populations, so there is a missing number.

p.11 top: ≥ 2 Mb – this is in a different font to the rest of the ms.

p.11 bottom & top half of page 12: "When estimating inbreeding levels based on ROH ≥ 2 Mb, we found that, on average, less than 10% of the genomes contain longer ROH segments (plus a similar statement earlier)." AND "These low levels of inbreeding and high genetic diversity imply that the two surviving populations have not yet been affected by strong inbreeding depression3."

The authors refer to inbreeding coefficients (F) of less than 10% as if they are benign. However, even inbreeding coefficients of at least 0.045 and 0.086 as they report (Supplementary Table 4) for the Bornean and Sumatran rhinoceros populations are expected to be having harmful effects on total fitness due to inbreeding depression. If we assume a conservative value of 6 haploid lethal equivalents (B) (O'Grady et al. 2006; Frankham et al. 2017, Chapter 3), we expect inbreeding depression (e-FB) of 24% and 40% in the Bornean and Sumatran populations, which are hardly benign.

I suggest that the wording be modified.

p.25: Reference 24 is now published in Evolution Letters

O'Grady, J.J., Brook, B.W., Reed, D.H., et al., 2006. Realistic levels of inbreeding depression strongly affect extinction risk in wild populations. *Biological Conservation* 133, 42-51.

Reviewer #3:

Remarks to the Author:

I think in general the authors have dealt with the concerns brought up in my review.

I would take some issue with this response however,

"We agree that addressing the taxonomy of Sumatran rhinoceros is interesting and important for conservation, but is however not the main focus of this manuscript. It is our view that in order to resolve the taxonomy properly, one would need genomes from additional extinct populations (such as Burma and China)."

I think the relevant question here in regards to species limits is whether the Borneo populations and those in Sumatra and Peninsular Malaysia represent two different, evolutionarily independent species lineages or not. Whether those extinct populations formerly in Burma, Indo-china and China (*D. s. lasiotis*) would also represent one or more independent lineages is a separate question. Those populations are not available to the ongoing conservation efforts. I think the sampling in this paper could be used to more directly test hypotheses regarding species limits between *D. s. sumatrensis* and *D. s. harrisoni* so that conservation efforts may be done clear-eyed in regards to how crossing the populations may or may not be creating species-hybrids. Again, given the precarious situation for these populations this may certainly be justified but I think we should be aware of the implications of these conservation decisions.

But, overall this is an excellent paper and I don't see this point of discussion as being a stumbling block to publication and look forward to seeing this and follow-up work in print.

REVIEWERS' COMMENTS

Reviewer #2 (Remarks to the Author):

Re-Review of Von Seth for Nature Communications

The authors have addressed my comments in a careful manner. In particular, the information on mutation load is much clearer and easier to follow.

>>> We thank the reviewer for their useful evaluation and are pleased to hear the manuscript is now easier to read.

However, I do have a few remaining comments as follows:

p.8 mid: In total, we found 373 LoF variants across all three populations, and found that Borneo has a significantly higher number of LoF variants compared to the other two modern populations (t-test, $p = 2.78e-05$), and that Sumatra has significantly higher number of LoF variants compared to Malay Peninsula (t-test, $p = 0.0019$, Whilst this is true, it is misleading as the text gives values of 219, 208 and 136 for B, S, and M populations. Thus, B & S >> M, but B is slightly > S. I suggest that re-wording is justified.

>>> Although we think that the reviewer had confused the numbers between Figure 3 c) and Supplementary Table 8, we agree that our writing here was not sufficiently clear. In figure 3 c) we present the number of LoF variants carried by each individual grouped by population, and in Supplementary Table 8 we have summed up the number of genes carrying one to six LoF variants. We have now added the population average numbers of LoF variants to this sentence in the hope of clarifying any confusion (see lines 250-253).

p.9 Temporal Changes... and elsewhere: “heterozygosity (θ , heterozygous sites/kb)” – this form of heterozygosity is now defined in many places (I believe in response to my previous comment): It only needs to be defined once if all cases of heterozygosity are defined in the same way (which I think is the case).

>>> We thank the reviewer for pointing this out, and have now removed the definition from all places except for the first time it is mentioned.

p.9 bottom-10 top: “Among these, there were 1, 382, and 762 genes that had fixed variants in the Sumatra, Borneo, and Malay Peninsula populations, respectively (Supplementary Table 6).”

There are two numbers and three populations, so there is a missing number.

>>> We wrote ‘1’ instead of ‘one’, and can see why the reviewer interpreted that as two numbers instead of three. However, we noticed that we had made a mistake when reporting these numbers, which are now: zero, 103 and 197. We have updated the numbers accordingly.

p.11 top: ≥ 2 Mb – this is in a different font to the rest of the ms.

>>> We thank the reviewer for noticing this and have changed the font to be the same as the rest of the ms.

p.11 bottom & top half of page 12: “When estimating inbreeding levels based on ROH ≥ 2 Mb, we found that, on average, less than 10% of the genomes contain longer ROH segments (plus a similar statement earlier).”AND “These low levels of inbreeding and high genetic diversity imply that the two surviving populations have not yet been affected by strong inbreeding depression³.”

The authors refer to inbreeding coefficients (F) of less than 10% as if they are benign. However, even inbreeding coefficients of at least 0.045 and 0.086 as they report (Supplementary Table 4) for the Bornean and Sumatran rhinoceros populations are expected to be having harmful effects on total fitness due to inbreeding depression. If we assume a conservative value of 6 haploid lethal equivalents (B) (O’Grady et al. 2006; Frankham et al. 2017, Chapter 3), we expect inbreeding depression (e-FB) of 24% and 40% in the Bornean and Sumatran populations, which are hardly benign.

I suggest that the wording be modified.

>>> The reviewer is correct that even a small number of lethal equivalents can cause inbreeding depression. We have added on lines 378-379 that “even a small number of lethal equivalents could in theory lead to reductions in fitness” while citing Frankham et al. 2017 and O’Grady et al. 2006.

We have also deleted “inbreeding is low” from the sentence on lines 407-410 so that it reads “For the extant Sumatran rhinoceros populations, since most of the identified LoF variants were in heterozygous state, gene flow would probably not lead to a marked increase in the masking of deleterious alleles.”

In the last paragraph, on line 471, we have changed “without evidence for recent inbreeding” to “with little evidence for recent inbreeding”.

Finally, to put the proportion of longer ROH segments in Borneo and Sumatra into a clearer context, we also write:

- on lines 370-372: “even though the Sumatran rhinoceros has gone through a major decline in the past century, to the extent that fewer than 100 individuals currently remain, we find relatively little evidence for recent inbreeding in the populations on Borneo and Sumatra.”
- on lines 385-387: “In contrast to the extant populations, the recently extinct population on the Malay Peninsula had lower mutational load, but a higher proportion of the genome contained within longer ROH segments (30%)”

p.25: Reference 24 is now published in Evolution Letters

>>> We thank the reviewer for pointing this out and have now updated that reference.

O’Grady, J.J., Brook, B.W., Reed, D.H., et al., 2006. Realistic levels of inbreeding depression strongly affect extinction risk in wild populations. Biological Conservation 133, 42-51.

Reviewer #3 (Remarks to the Author):

I think in general the authors have dealt with the concerns brought up in my review.

I would take some issue with this response however,

"We agree that addressing the taxonomy of Sumatran rhinoceros is interesting and important for conservation, but is however not the main focus of this manuscript. It is our view that in order to resolve the taxonomy properly, one would need genomes from additional extinct populations (such as Burma and China)."

I think the relevant question here in regards to species limits is whether the Borneo populations and those in Sumatra and Peninsular Malaysia represent two different, evolutionarily independent species lineages or not. Whether those extinct populations formerly in Burma, Indo-china and China (*D. s. lasiotis*) would also represent one or more independent lineages is a separate question. Those populations are not available to the ongoing conservation efforts. I think the sampling in this paper could be used to more directly test hypotheses regarding species limits between *D. s. sumatrensis* and *D. s. harrisoni* so that conservation efforts may be done clear-eyed in regards to how crossing the populations may or may not be creating species-hybrids. Again, given the precarious situation for these populations this may certainly be justified but I think we should be aware of the implications of these conservation decisions.

But, overall this is an excellent paper and I don't see this point of discussion as being a stumbling block to publication and look forward to seeing this and follow-up work in print.

>>> We sincerely thank the reviewer for their positive evaluation and important discussion points, and we will consider these thoughts for a follow-up paper on Sumatran rhinoceros population divergence and phylogeny.